# Handlebar Width Choices Must Be Considered for Female Cyclists

**DOI:** 10.3390/jfmk10010028

**Published:** 2025-01-10

**Authors:** Zi-Jun Lin, Pei-Chen Tsai, Chia-Hsiang Chen

**Affiliations:** 1Graduate Institute of Athletics and Coaching Science, National Taiwan Sport University, Taoyuan 333325, Taiwan; warmupfl2@gmail.com; 2Office Physical Education, National Pingtung University of Science and Technology, Pingtung 912301, Taiwan; pchentsai@gmail.com

**Keywords:** bike, fitting, EMG, motion analysis, female athletes

## Abstract

**Background**: The effects of handlebar width on female cyclists are understudied; therefore, it is necessary to find an optimal handlebar width for women based on anatomical features. **Methods**: Ten healthy women participants whose muscle activity and movements were measured using four kinds of handlebar widths were observed using EMG and 3D motion analysis systems. Participants cycled at a constant cadence and power output using different handlebar widths in a counterbalanced order. The kinematic results and muscle activation, as a consequence of using different handlebar widths, were compared using a one-way repeated measures ANOVA (α = 0.05). **Results**: It was discovered that using a medium-width handlebar not only resulted in significantly lower bicep activation compared to narrow and self-selected widths, but also resulted in less triceps and latissimus dorsi activation compared to the self-selected width. Regarding kinematics, using a medium-width handlebar significantly reduced hip ROM, while using a narrow handlebar led to greater hip adduction. **Conclusions**: Cyclists are advised to use a handlebar width that matches their shoulder width, since this may avoid muscle fatigue while also allowing for better hip posture. However, commercial models are usually wider than female shoulders. Thus, these results provide insights useful for future handlebar design.

## 1. Introduction

In recent years, gender differences have become an increasingly important focus in cycling-related research [1,2]. Although individuals of comparable body sizes may appear similar, males typically exhibit greater skeletal dimensions and bone mass than females [3]. Moreover, males are reported to have a higher overall injury incidence, while females demonstrate a higher prevalence of overuse injuries [4]. These distinctions have a direct impact on bicycle fitting strategies and user experiences. The three parts of the bicycle that come into contact with the human body are the handlebar, the seat, and the pedals, and changes among these three parts are inextricably linked. Many studies have attempted to apply exercise biomechanics to identify the best setting for adjusting cycling posture, with the aim of improving exercise performance, increasing comfort, and decreasing injuries [5,6,7,8,9,10]. However, current studies on bicycle posture have all been conducted using male subjects, with very few studies targeting women, except for studies on bicycle seats. Past studies have shown that there are differences in sit bone width between men and women, indicating varying requirements for bicycle saddle width [11]. Additionally, studies have shown that the mean shoulder width of women and men are 32.9 ± 1.5 cm and 38.1 ± 1.8 cm, respectively [12], but the current handlebar width of commercially available bicycles is 36–44 cm. This implies that the current bicycle handlebar width is not suitable for cyclists with narrow shoulders and smaller bodies. Therefore, to improve posture adjustment methods for cyclists, it is essential to investigate the possible effects of handlebar width.

Numerous bike-fitting professionals, researchers, and clinicians have access to diverse technologies that aid in refining cycling positions and identifying components tailored to individual needs [13]. Ironically, however, existing equipment for posture adjustments appears to be unsuitable for female cyclists. Neither the design nor configuration of road bicycles are suitable for women, who are generally shorter in stature than men [1]. Men and women have different anatomical and biomechanical considerations, and past posture adjustment methods were developed based on the male anatomy [2]. Previous studies found that women have more knee abduction and knee external rotation angles during exercise than men [14] and were 2.23 times more likely to develop patellofemoral pain syndrome [15]. Furthermore, the incidence of sports injuries is higher in female cyclists than in their male counterparts, which may also be due to differences in body structure [16,17]. In summary, we can conclude that improving cycling posture in women is an important issue that cannot be ignored. From the perspective of cycling posture adjustment, handlebars are as important as seats and pedals, but there is insufficient evidence regarding this issue in female cyclists.

Previous studies have indicated that the handlebar width should be approximately equal to the width of the shoulders [18]. Incorrect handlebar position may affect upper-limb muscle recruitment, resulting in fatigue and pain, while also increasing the susceptibility of the palms, elbow joints, and shoulder joints to pain and discomfort [19]. Wide handlebars, for instance, may result in strains on the trapezius and rhomboid muscles [20]. In fact, the shoulders are the most common site of overuse-related fatigue and pain in cycling [21]. Researchers have noted that fatigue may impair shoulder strength, proprioception, and range of motion, all of which are potential risk factors for shoulder overuse injuries [22,23]. This is partly due to excessive elbow extension, which can decrease shock absorption [24]. During cycling, the degree of muscle activation increases in the biceps, deltoids, and latissimus dorsi as power output increases. This is because left and right pedaling causes the center of gravity to shift, meaning more upper-limb muscles need to be recruited to maintain trunk stability and support upper-body weight [25]. In addition, during high-intensity sprint cycling, using wider handlebars may increase the degree of lateral displacement of the center of gravity, while using narrower handlebars may result in difficulty maintaining the balance of the bicycle. Therefore, an appropriate handlebar width could help cyclists control the bicycle and exchange forces, particularly at high-intensity exercise. Taken together, these studies demonstrate the importance of handlebar position. However, there are currently no studies on whether handlebar width affects muscle activation. Previous studies have shown that different hand widths affect push-ups and different grip widths affect bench press performance [26,27]. Even though cycling does not require significant involvement of the upper limbs, the load produced from long periods of cycling is notable. Therefore, using unsuitable handlebars may increase upper-limb load and hence increase the incidence of injuries.

It is worth noting that the biomechanics of the lower limbs are also associated with handlebar adjustments. Excessive forward or lowering handlebar movements will increase the lumbar flexion angle in cyclists, making them more prone to lower-back pain [28,29]. In addition, adopting a dropped posture during cycling decreases the angle between the legs and trunk, which results in more efficient hip extension, while also decreasing muscle activation and joint stress [30]. Consequently, changes in handlebar position affect upper-body posture, which in turn has implications for the lower body. Moreover, using different seats alters the pelvic tilt angle and consequently affects shoulder and elbow angles [31]. Therefore, handlebars are an essential component for adjusting posture. However, existing research has only explored handlebar positions, with no study verifying whether the current handlebar width standards are suitable for individuals with larger or smaller body proportions. Therefore, this study aims to investigate how handlebar width influences muscle activation patterns in the upper body and affects the kinematics of the lower-limb joints during cycling. Specifically, a wider grip may decrease the trunk angle by positioning the arms further outward, while a narrower grip may increase the trunk angle by bringing the arms closer to the body. These adjustments in upper body posture could, in turn, alter the angles of lower-limb joint movements. Based on this rationale, we hypothesize that using handlebars that match the shoulder width will reduce muscle activation in the upper body and lead to changes in lower-limb joint angles compared to handlebars that are either wider or narrower than shoulder width.

## 2. Materials and Methods

### 2.1. Participants

The objective of this study was to investigate individuals with smaller body proportions. To this end, this study ultimately recruited 10 female participants. The characteristics of the participants were as follows: average age, 20.6 ± 1.4 years; height, 162.1 ± 4.8 cm; weight, 57.2 ± 7.8 kg; and shoulder width, 33.2 ± 1.3 cm. All participants had engaged in regular workout routines for at least one year (a minimum of 600 h per year). None had experienced musculoskeletal or neuromuscular injuries in the lower limbs within six months prior to this study. Additionally, this study recruited participants for this cross-sectional study whose shoulder widths matched the tested handlebar widths (30, 32, or 34 cm). Shoulder widths of 28 and 36 cm were considered the extremes in this study, and handlebars outside this range were not considered. In a power analysis for a repeated measures analysis of variance, G*Power 3.1.9.7 was used to calculate the appropriate sample size for this study. Using a power of 0.80, an α of 0.05, and an effect size of 0.20, the required sample size was determined to be 10, which met the power criterion [32]. The participants provided informed consent prior to the experiment. This study was approved by the National Cheng Kung University Human Research Ethics Committee (IRB #: NCKU HREC-E108-028-2) and conducted in accordance with the principles of the World Medical Association Declaration of Helsinki.

### 2.2. Equipment

This study was conducted in a sports biomechanical laboratory. Muscle activation was measured using a wireless electromyography (EMG) system (Trigno Wireless System, Delsys Inc., Natick, MA, USA). Surface EMG signals were used to measure surface myoelectrical activity [33,34,35], and the sampling frequency was 1000 Hz. Sensors were attached to the biceps brachii (BB), triceps brachii (TB), deltoid (DEL), latissimus dorsi (LAT), rectus femoris (RF), biceps femoris (BF), tibialis anterior (TA), and gastrocnemius (GAS). Before positioning the electrodes, the contact area was shaved and cleaned using alcohol wipes to eliminate possible interruption on the skin surface. A three-dimensional motion analysis system (Cortex 7.1; Motion Analysis Corp., Santa Rosa, CA, USA) was employed [36,37], and four infrared cameras (Kestrel 2200; Motion Analysis Corp., Santa Rosa, CA, USA) were used to acquire the three-dimensional trajectory of reflective points. The sampling frequency was 200 Hz, referencing past studies on cycling [5,38], and the built-in module of the motion analysis system, the 6-degrees-of-freedom (DOF) Helen Hayes Marker Model (Figure 1) [39], was used. The rotation sequence for the joints was flexion/extension, abduction/adduction, and internal/external rotation for the hip, knee, and ankle. This was used to capture the participants’ movement patterns during cycling [5,40]. Due to the possibility that the shortest handlebar width of commercially available bicycles, which is 36 cm, may not be suitable for individuals with smaller body proportions, the handlebar width was designed with reductions of two centimeters each. Five handlebar sizes were used in this experiment: 28, 30, 32, 34, and 36 cm. All handlebars were precut to length at a factory before welding. The material used was an aluminum alloy (Figure 2).

### 2.3. Protocols

In this experiment, a moderate handlebar width was defined based on the participant’s shoulder width (length between the two shoulder peaks), whereas broad and narrow handlebars were defined by adding or subtracting 2 cm from the shoulder width, respectively. The participants also chose a handlebar based on personal preference. To reduce experimental errors, all tests were conducted using the same bike (Giant Propel Advanced 2). Moreover, all participants were tested during the luteal phase of their menstrual cycle to minimize potential variability caused by hormonal fluctuations, as the menstrual cycle can influence pain perception and athletic performance [41,42,43]. To ensure consistency and minimize the influence of individual body segment differences among participants, the cycling posture was standardized with the handlebar width matching the participant’s shoulder width, a knee flexion angle of 30 degrees, and a trunk flexion angle of 45 degrees when the crank was in the 6 o’clock position [8,44]. The experiment was conducted under single blind conditions, and the handlebars were mounted by the same practitioner to minimize potential bias. Prior to the experiment, environmental calibration was performed, and the experimental setup is illustrated in Figure 1. Additionally, a maximal voluntary contraction (MVC) test was conducted for standardization before the start of the experiment (Table 1). Following a 10 min dynamic warm-up, the experimental procedures commenced. The participants commenced the experimental trials in a counterbalanced order to minimize random errors caused by adaptation. Each trial comprised 10 min of cycling. To prevent the participants from deliberately manipulating pedaling techniques, they were not informed when data were being recorded. The recording lasted for 30 s [11,45,46], during which at least 45 pedal cycles were recorded. Between each trial, the participants engaged in passive resting for 5 min. During the trials, the participants pedaled at a constant cadence and power output (90 rpms/100 watts), which were controlled through visual feedback from the power meter (Power Pro; Giant, Taichung, Taiwan). Considering that the participants in this study were general cycling enthusiasts, generating 100 watts is relatively easy for healthy adults, and this pedaling power can prevent participants from experiencing lower-limb fatigue during the experiment, which could otherwise affect the experimental results. Experienced cyclists typically prefer a higher cadence, around 90 RPM or higher [47,48].

### 2.4. Data Analysis

Cortex three-dimensional motion analysis software (Cortex 7.1, Motion Analysis Corporation, Santa Rosa, CA, USA) was used to analyze lower-limb kinematics during cycling. A Butterworth fourth-order low-pass filter was used to remove high-frequency noise in the selected data, and the cutoff frequency was set at 6 Hz. Each participant’s general information (height, weight, and limb segment parameters) was inputted to construct a computational human body model. The body segments were modeled as rigid bodies, and relative angles were obtained using the joint center as a fixed point. The kinematic changes in the hip, knee, and ankle joints during cycling were calculated. In addition, Delsys EMGwork 4.5.4 Analysis software was used to analyze the myoelectrical signals that were collected. The Butterworth fourth-order filter was used to perform 10−500 Hz band-pass filtering, followed by full-wave rectification and smoothing with a low-pass filter (6 Hz). Finally, the degree of muscle activation was calculated.

### 2.5. Statistics

Analyses were performed using IBM SPSS Statistics for Windows, version 22 (IBM Corp., Armonk, NY, USA), and the results were expressed as means ± SDs. To analyze the differences in kinematics and kinetics according to different handlebar widths used for cycling, a one-way repeated measures analysis of variance (ANOVA) was performed, with handlebar width (narrow, moderate, broad, self-chosen) as the independent variable, and kinematics (hip, knee, ankle) and kinetics (BB, TB, DEL, LAT, RF, BF, TA, and GAS) as the dependent variables. When the main effect size was significant, the Bonferroni method was used for post hoc comparison. Statistical significance was set at *p* ≤ 0.05. The value of the confidence interval was set at 95%. The effect size was determined according to the square value of net correlation Eta (η^2^). The degree of effect size was defined as follows: 0.01–0.06 = small, 0.06–0.14 = moderate, and >0.14 = large [49].

## 3. Results

Table 1 illustrates muscle activation resulting from the use of different handlebar widths. Based on the ANOVA, a significant difference in muscle activation was observed for the BB, TB, and LAT, whereas no significant difference was found for the DEL, RF, BF, TA, and GAS.

Table 2 shows the lower-limb joint angle of participants when handlebars of different widths were used for cycling. A one-way repeated measures ANOVA revealed significant differences in hip flexion/extension range of motion (ROM). There were no significant differences in other lower-limb kinematic parameters.

Post hoc comparisons among different handlebar widths showed that the use of the moderate handlebar was associated with the following: lower BB activation compared with that associated with the narrow and self-chosen handlebars, lower TB activation compared with that associated with the self-chosen handlebar, and lower LAT activation than that associated with the broad and self-chosen handlebars (Table 2). Hip flexion/extension ROM was less with the moderate handlebar than with the self-chosen handlebar, while hip adduction was greater with the narrow handlebar than with the moderate handlebar (Table 3).

## 4. Discussion

The purpose of this study was to examine the effects of different handlebar widths (narrow, moderate, broad, self-chosen) during cycling on muscle activation and lower-limb kinematics. Our findings revealed that changing handlebar widths not only affected the degree of muscle activation, but also altered hip ROM, which supports the hypothesis of this study. Female cyclists who used moderate handlebars had lower BB and LAT activation and smaller hip ROM and adduction. Therefore, using moderate handlebars may enhance comfort during long periods of cycling.

The results of this study indicate that different handlebar positions affect the degree of upper-limb muscle activation, which was consistent with findings of previous studies [44]. However, previous studies mainly focused on handlebar height, and a lower handlebar position will cause the cyclist’s center of gravity to tilt forward, thus increasing the pressure on the seat and hence the risk of sports injuries [50]. However, very few studies have examined the effects of handlebar width on female cyclists. Our study found that the mean shoulder width of participants was 33.25 ± 1.39 cm, which is a standard shoulder width in women (32.9 ± 1.5 cm), as reported previously [12]. Existing studies recommend that optimal handlebar width should be the same as shoulder width [8,24]. This study also found that using a handlebar with the same width as the shoulders of female cyclists decreased upper-limb muscle activation, which may decrease discomfort in women during long periods of cycling. However, the current handlebar widths of commercially available bicycles range from 36 cm to 44 cm. Therefore, according to our study results, current handlebars do not meet the needs of most female cyclists, and more than 50% require smaller handlebars, which would allow them to select the best handlebar position. There are known differences in body composition between males and females [51]. In terms of both absolute values and relative body weight, males have greater muscle mass than females, with this difference being more pronounced in the upper body [52,53]. Additionally, studies have shown that at the same saddle height, females exhibit greater knee extension angles compared to males [54], which may be partially attributed to gender differences in pelvic structure [55]. Therefore, even with the same height, gender can influence cycling performance. In conclusion, selecting a handlebar width equal to shoulder width serves as a starting point for female bike fitting, but is not always directly related to cycling efficiency.

This study also found that changing handlebar widths in female cyclists did not affect lower-limb muscle activation or changes in knee and ankle joint angles, but did affect changes in hip joint angle. Previous studies found that using handlebars of different widths would not affect lower-limb muscle activation and pedaling efficiency [19]. Another study highlighted that although using different handlebar heights and front/rear positions for cycling did not affect pedaling efficiency or changes in knee and ankle joint angles, it did alter the trunk angle, and hence affected the hip flexion angle (70–90°) [6]. Therefore, selecting handlebars of any size will not affect lower-limb movement and pedaling efficiency during cycling. However, improper handlebar positioning will increase hand, ankle, and back discomfort, and increase physiological burden [56]. In this study, selecting handlebars of any size did not affect lower-limb movement and pedaling efficiency during cycling. This may be because changing handlebar size does not impact riding posture as significantly as adjusting handlebar position, but it can improve the activation levels of upper-limb muscles. During cycling, a higher pedaling power output will lead to greater anterior pelvic tilt, causing the center of gravity to shift forward and increasing hand pressure [57]. However, in this study, the pedaling power was relatively low, and compared to high-intensity cycling, the involvement of the upper body was limited. Therefore, the effects of handlebar changes may be smaller during low-intensity cycling, which is a limitation of this study. However, the anterior pelvic tilt of women is greater than that of men [57], and hence female cyclists are more prone to pelvic tilt associated with changes in handlebar position than male cyclists [50,57]. These changes cause a greater shift in body weight from the sit bones to the perineum, which may increase discomfort during long periods of cycling and even increase the incidence of sports injuries.

The results of this study showed that using a moderate handlebar width led to lower BB and LAT activation and lower hip ROM compared with those associated with self-chosen handlebar widths. Although these findings do not impact pedaling efficiency, changes in the hip joints are closely associated with lower back pain [58]. In addition, cyclists usually need to maintain a fixed posture for long periods of cycling, and hence the incidence of injury may increase with time. Sports science can be applied to achieve a more accurate selection of handlebars suited to different cyclists. This not only helps to decrease upper-limb muscle load, but also reduces hip ROM, which may decrease discomfort in women caused by long periods of cycling. Therefore, selecting a handlebar width that matches the shoulder width is an extremely important step in bike fitting, particularly for smaller-bodied individuals and women, who often have narrower shoulders and may benefit more from optimized handlebar widths. While this recommendation appears most applicable to these groups, further studies are needed to explore its relevance to other populations, such as men or individuals with broader shoulders, as well as its implications for different cycling styles and conditions.

Based on the findings, it can be hypothesized that the observed reduction in muscle activation and hip ROM with moderate handlebar widths may be attributable to optimized biomechanical alignment, which could potentially reduce muscular fatigue and joint strain during prolonged cycling. Future studies could investigate whether similar patterns of muscle activation and hip ROM changes are observed in other populations, such as male cyclists or individuals with varying shoulder widths, to establish a more generalizable framework for bike fitting recommendations.

Finally, while the current trend in professional cycling emphasizes narrower handlebars due to aerodynamic benefits [59], this study was conducted in a controlled laboratory setting, which did not account for aerodynamic factors or simulate the dynamic conditions of outdoor cycling on varied terrains. Therefore, the findings of this study should only be considered as a reference for recreational cyclists.

## 5. Conclusions

We recommend that female cyclists focus on selecting handlebars with the same width as their shoulders to achieve better pedaling efficiency. However, at present, there are no commercially available handlebar widths that are less than 36 cm. Therefore, we hope that bicycle manufacturers can better meet the needs of women in the future.

## Figures and Tables

**Figure 1 jfmk-10-00028-f001:**
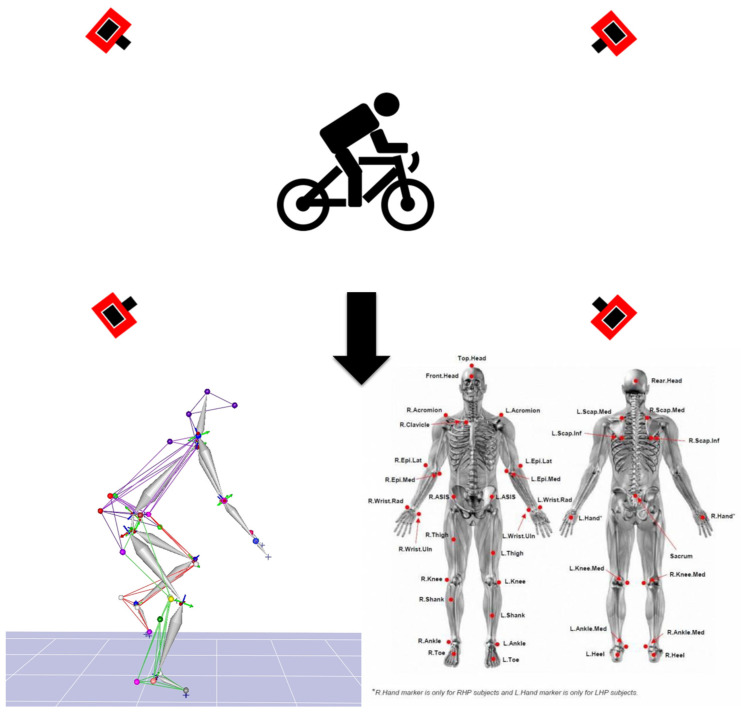
Experimental Setup Diagram.

**Figure 2 jfmk-10-00028-f002:**
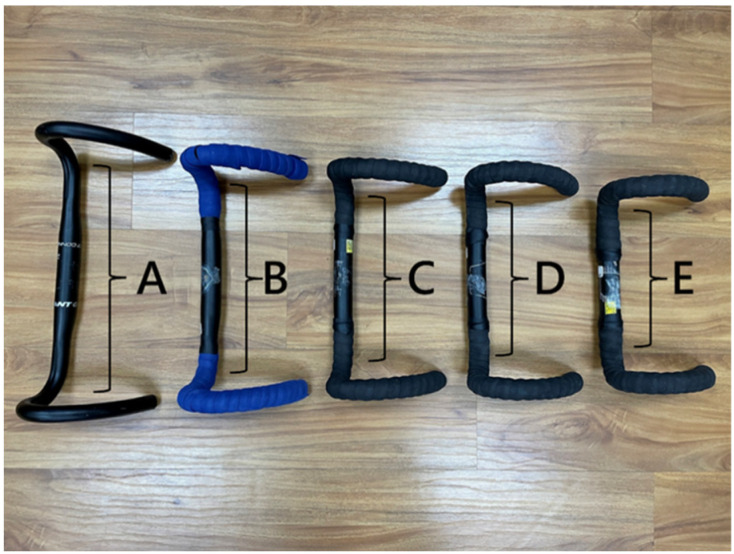
There were five types of handlebars used in this study according to their widths: 36 cm (A), 34 cm (B), 32 cm (C), 30 cm (D), and 28 cm (E).

**Table 1 jfmk-10-00028-t001:** Procedure to perform the maximal voluntary contraction (MVIC) of each muscle.

Muscle Group	Description	
Biceps brachii (BB)	Perform elbow flexion by contracting the biceps brachii and bending the arms upward, while simultaneously applying manual resistance.	** 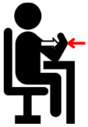 **
Triceps brachii (TB)	Perform elbow extension by straightening the arms to their maximum extent, while simultaneously applying manual resistance.	** 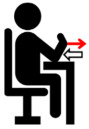 **
Deltoid (DEL)	Lift the shoulders and upper arms until they form an angle of approximately 60 to 90 degrees with the ground, while simultaneously applying manual resistance.	** 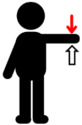 **
Latissimus dorsi (LAT)	Perform elbow flexion and push backward while simultaneously applying manual resistance.	** 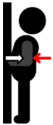 **
Rectus femoris (RF)	Extend the knee and apply manual resistance simultaneously.	** 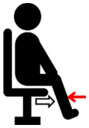 **
Biceps femoris (BF)	Bend the knee, bringing the thigh toward the buttocks, while simultaneously applying manual resistance.	** 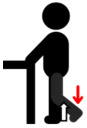 **
Tibialis anterior (TA)	Lift the ankle upward while simultaneously applying manual resistance.	** 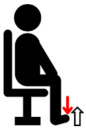 **
Gastrocnemius (GAS)	Push down on the toes to raise the heels while simultaneously applying manual resistance above the knees.	** 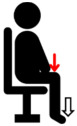 **

The red arrow represents the direction of resistance, while the white arrow represents the direction of muscle contraction.

**Table 2 jfmk-10-00028-t002:** Differences in muscle activation when riding with different handlebar widths.

MVC%	Narrow	Moderate	Wide	Self-Chosen	*p*-Value	Effect Size (η^2^)
BB	32.74 ± 6.22 ^a^	22.66 ± 8.77 ^a,b^	22.84 ± 8.96	31.04 ± 9.55 ^b^	*p* = 0.002 *	0.40
TB	36.44 ± 3.13 ^c^	38.76 ± 3.11	36.59 ± 4.85	41.30 ± 3.65 ^c^	*p* = 0.006 *	0.36
DEL	30.62 ± 9.83	24.00 ± 9.35	29.20 ± 4.51	28.42 ± 7.85	*p* = 0.113	0.19
LAT	30.88 ± 3.19	27.89 ± 4.63 ^b,d^	33.54 ± 2.45 ^d^	32.71 ± 2.71 ^b^	*p* = 0.000 *	0.53
RF	26.52 ± 4.20	25.67 ± 7.56	24.99 ± 3.34	26.28 ± 2.51	*p* = 0.811	0.03
BF	24.87 ± 6.04	20.10 ± 6.94	23.77 ± 2.77	25.60 ± 4.05	*p* = 0.122	0.19
TA	34.00 ± 1.62	34.52 ± 1.73	34.02 ± 1.35	33.82 ± 2.20	*p* = 0.505	0.08
GAS	34.56 ± 0.86	34.78 ± 1.73	35.34 ± 1.13	35.36 ± 1.00	*p* = 0.255	0.13

* Indicates significant differences. ^a^ Significant difference with handlebar widths of narrow and moderate size. ^b^ Significant difference with handlebar widths of moderate and self-chosen size. ^c^ Significant difference with handlebar widths of narrow and self-chosen size. ^d^ Significant difference with handlebar widths of moderate and wide size.

**Table 3 jfmk-10-00028-t003:** Differences in lower-limb joint angles during cycling for different handlebar widths are shown in the table for ranges of motion in all three axes (sagittal, frontal, and transverse planes).

Degrees (+/−°)	Narrow	Moderate	Wide	Self-Chosen	*p*-Value	Effect Size (η^2^)
Hip FE/EXT rom	54.29 ± 8.07	48.71 ± 2.98 ^b^	51.68 ± 5.79	52.03 ± 5.01 ^b^	*p* = 0.008 *	0.35
Hip ADD/ABD rom	7.93 ± 3.57	5.71 ± 1.56	7.67 ± 4.84	8.30 ± 3.47	*p* = 0.067	0.23
Hip IR/ER rom	50.66 ± 33.26	50.41 ± 28.84	49.54 ± 25.69	51.61 ± 32.22	*p* = 0.968	0.02
Knee FE/EXT rom	89.77 ± 28.91	80.64 ± 11.36	83.97 ± 19.48	86.18 ± 17.36	*p* = 0.344	0.11
Knee ADD/ABD rom	42.94 ± 29.72	43.17 ± 26.63	42.53 ± 26.31	42.40 ± 31.31	*p* = 0.993	0.00
Knee IR/ER rom	36.10 ± 33.51	26.36 ± 12.85	30.24 ± 20.40	41.76 ± 30.05	*p* = 0.301	0.12
Ankle DF/PF rom	27.56 ± 7.89	27.03 ± 4.02	25.89 ± 7.06	35.67 ± 31.66	*p* = 0.435	0.09
Ankle ADD/ABD rom	8.88 ± 3.94	8.73 ± 2.85	9.66 ± 5.90	14.23 ± 17.71	*p* = 0.470	0.08
Ankle IR/ER rom	7.57 ± 2.41	6.90 ± 1.94	11.68 ± 10.79	15.15 ± 18.81	*p* = 0.138	0.18

* Indicates significant differences. ^b^ Significant difference with handlebar widths of moderate and self-chosen size. FE = Flexion, EXT = extension, ADD = adduction, ABD = abduction, IR = internal rotation, ER = external rotation, PF = plantarflexion, DF = dorsiflexion, and rom = range of motion.

## Data Availability

The data are not publicly available due to privacy or ethical restrictions, but can be obtained from the corresponding author upon reasonable request.

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
