# Peer review of "Handlebar Width Choices Must Be Considered for Female Cyclists"

_jfmk, 2025, doi:10.3390/jfmk10010028_

Round 1
Reviewer 1 Report
Comments and Suggestions for Authors
Dear Authors,
First and foremost, congratulations on submitting your paper for publication. Anatomical, biomechanical and functional sex differences represents a current topic that needs further investigation. After reviewing your paper, here are a few suggestions to improve your paper.
INTRODUCTION
- - Second paragraph, from line 28 to line 41, please add more references that highlighted sex differences, in order to support your choice.
- - In the third paragraph, from line 57 to line 77, the concept of fatigue in the shoulder and their assessment should be better described in order to support your work. In fact, shoulder is an area where fatigue and pain could commonly occur. Researchers have suggested that fatigue might affect the shoulder’s strength, proprioception, and range of motion, representing possible risk factors for overuse shoulder injury. Please take in to consideration these articles:
Buoite Stella A, Cargnel A, Raffini A, Mazzari L, Martini M, Ajčević M, Accardo A, Deodato M, Murena L. Shoulder Tensiomyography and Isometric Strength in Swimmers Before and After a Fatiguing Protocol. J Athl Train. 2024 Jul 1;59(7):738-744. doi: 10.4085/1062-6050-0265.23. PMID: 38014804; PMCID: PMC11277270.
Côté JN. Adaptations to neck/shoulder fatigue and injuries. Adv Exp Med Biol. 2014;826:205-28. doi: 10.1007/978-1-4939-1338-1_13. PMID: 25330893.
- - The introduction would benefit from a clearer statement of the study's objectives and hypotheses. Explicitly defining these early on would enhance the reader's understanding of the study's direction.
METHOD
- -The characteristics of your sample should be described in your results section, not here. In method section you should describe inclusion and exclusion criteria
- - Please add more informations regarding Institutional Review Board such as name, number and date.
- - 2.2. Equipment. Please add more references that support the assessment tools used.
- -Considering that you wanted to evaluate women, it is important to specify at what stage of the menstrual cycle they were evaluated. In fact, the menstrual cycle influences both pain and sports performance.
- -The manuscript does not clearly justify the chosen sample size.
RESULTS
- - Please described here the characteristics of your sample.
- -How many women were included?
DISCUSSION
- -More thorough consideration of the study's generalizability would help readers assess the findings' applicability
- -The discussion would benefit from a comprehensive discussion of the study's limitations that could provide a more balanced view.
- -A suggestions of general hypotheses concerning the degree of muscle activation and the hip range of motion should be speculated.
- -Sex differences respect to others study on male should be presented
Reviewer 2 Report
Comments and Suggestions for Authors
As a person who rides a bicycle, every new article related to this subject piques my interest, nevertheless, the reviewed work raises questions that need answers. Most people who take cycling seriously have undergone the process of bike fitting in which the issue of handlebar width is one of the important although, in my opinion, not the primary issue. Below is a list of questions that arise after reading the paper. They mainly relate to the methodology used.
1. A questionable group with significant variation in body proportions (no analysis of limb proportions, position on the bike and torso position is a function of the length of the upper limbs of the torso and shoulder width) Failure to determine the length of the upper limbs means that the angle of the torso does not inform the position and angle of the upper limb.
2. Did each of the women surveyed complete each trial on the same bicycle or did they change to a new one with different sized handlebars? To what extent were the bicycles adjusted to each subject's morphological parameters (saddle height, saddle position in the front-rear plane, crank length, stem length, etc.)?
3. The setting of 45 degrees of flexion in the hips at what width of the handlebars was determined? As I presume it changed for each width of it. If, on the other hand, it was constantly maintained then by what method? By shortening the stem? Moving the saddle?
4. What was the grip of the handlebars during the trial? Not a clear description of the conditions of the experiment. It is difficult to imagine the conditions (mainly the position, the size of the bike, its individual setting for each athlete). If there was a grip, bottom? If so, what was the drop and rich handlebars? Was it the same steering wheel model in different sizes? The width of the handlebars is not its only parameter.
5. What width of the handlebars was considered moderate? Probably it was not possible to select a width equal to the individual shoulder width every time?
Results
6. Table 2 lacks a description of the significance of the d difference in Anova results.
Conclusions
7. What was the effect of the handlebars on the hip rom? Why was it smallest with moderate handlebar width? No explanation. Perhaps it is difficult to find a cause and effect relationship?
8. Is the data obtained a response to the need to maximize efficiency or comfort?
9. Handlebar width equal to shoulder width is the starting point in bikefitting, regardless of gender. But it is only a starting point. It is not always related to the efficiency of the cyclist's movement. The best evidence of this is the current trends in pro tour, where handlebars are being radically narrowed. Do the recommendations presented apply to female athletes or recreational cyclists? Biomechanical analyses and 3d models used in bikefit are enriched by analyzing the change in position of the cyclist's body related to the change in slope of the route. Estimating a 30-second activity for an activity that sometimes lasts for hours and is mostly implemented in varied terrain is a far-fetched approximation. Would the Authors apply an analogous recommendation for a 30-second attempt to pedal a route with 10% elevation or with a power output of the order of 5W/kg?
I will be obliged to clarify my doubts.
Round 2
Reviewer 1 Report
Comments and Suggestions for Authors
I thank the authors for the revision work done